# Factors Influencing the Distribution of Invasive Hybrid (*Myriophyllum Spicatum x M. Sibiricum*) Watermilfoil and Parental Taxa in Minnesota

**Jasmine A. Eltawely [1], Raymond M. Newman [1],*** 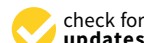 **and Ryan A. Thum [2]**

[1]  Department of Fisheries, Wildlife and Conservation Biology, University of Minnesota, Saint Paul, MN 55108, USA; eltaw003@umn.edu

[2]  Department of Plant Science and Plant Pathology, Montana State University, Bozeman, MT 59717, USA; ryan.thum@montana.edu

*   Correspondence: Rnewman@umn.edu

**Abstract:** Eurasian watermilfoil (*Myriophyllum spicatum* L.) hybridizes with the native northern watermilfoil (*M. sibiricum* Kom.), which raises new issues regarding management strategies to control infestations. To determine the distribution of hybrid (and coincidentally Eurasian and northern) watermilfoil in Minnesota, we sampled lakes across the state during 2017–2018 for watermilfoil. A total of 62 lakes were sampled, spanning a range of sizes and duration of invasion. Forty-three lakes contained Eurasian, 28 contained hybrid and 21 contained northern watermilfoil. Eurasian watermilfoil populations were widespread throughout the state. Hybrid populations were more commonly found in lakes in the seven county Twin Cities Metro and northern watermilfoil populations were more commonly found in lakes outside of the Metro area. We found no evidence that hybrid watermilfoil occurred in lakes environmentally different than those with Eurasian and northern watermilfoil, suggesting that hybrid watermilfoil is not associated with a unique niche. Hybrid watermilfoil presence was significantly associated with the Metro area, which may likely be due to spatial and temporal factors associated with hybrid formation and spread. Hybrid watermilfoil presence was also significantly associated with lakes that had more parking spaces and older infestations, but this relationship was not significant when the effect of region was considered. Hybrid watermilfoil populations were the result of both in situ hybridization and clonal spread and continued assessment is needed to determine if particularly invasive or herbicide-resistant genotypes develop.

**Keywords:** biological invasions; invasive plants; *Myriophyllum spicatum*; *Myriophyllum sibiricum*; hybridization; population genetics

## 1. Introduction

There is an increasing appreciation for the role of genetics and hybridization in invasion biology [1–5]. Hybridization between introduced and native species can lead to novel combinations of traits, novel trait values, or increased genetic variation, which may result in superior competitive phenotypes [2,3,6]. Hybrid species may thus have increased likelihood of survival and establishment success in novel habitats. Although most research on invasive hybrids has focused on terrestrial and wetland herbaceous plants [5,7], hybrids have been associated with invasiveness in woody plants [8], submersed aquatic plants [3,9] and animals [10]. Therefore, identifying hybrids and the factors associated with their occurrence and spread is important to understanding biological invasions and informing management of invasive species.

Eurasian watermilfoil (*Myriophyllum spicatum* L.) is one of the most heavily managed invasive aquatic plants in North America. It is native to Europe, Asia and Africa [11] and is now present in 48 states and three Canadian provinces [12]. Eurasian watermilfoil was first documented in

Minnesota in Lake Minnetonka in 1987 and White Bear Lake in 1988 and has since spread to more than 300 waterbodies [13]. Eurasian watermilfoil forms dense canopies that can reduce native species richness through the suppression of native vegetation [14]. As a result of the decrease in native plant populations, Eurasian watermilfoil can negatively affect animals that depend on the health of aquatic ecosystems [15]. Nuisance growth of Eurasian watermilfoil can also inhibit recreational use of waterbodies [16]. Millions are spent in the U.S. annually on the control of Eurasian watermilfoil and it is the most commonly managed invasive aquatic plant [15,17]. Despite widespread control and prevention efforts, new cases of Eurasian watermilfoil invasions accumulate every year (http://www.dnr.state.mn.us/invasives/ais/infested.html).

Eurasian watermilfoil (*Myriophyllum spicatum*) hybridizes with the native northern watermilfoil (*M. sibiricum* Kom.) [3] and concern has arisen that hybrid watermilfoil may respond differently to management or be more invasive than pure Eurasian [18,19]. Several studies indicate that some hybrid watermilfoil genotypes are less affected by certain commonly-used herbicides than Eurasian, including auxinic herbicides such as triclopyr and 2,4-D (2,4-dichlorophenoxy acetic acid) [18,20–22] as well as fluridone [23–26]. Parks et al. [21] found a greater reduction in Eurasian watermilfoil in comparison to hybrid, following treatment with auxinic herbicides, and similar results were found by Nault et al. [20] following treatment with 2,4-D. Fluridone-resistant populations of hybrid watermilfoil have been confirmed in several studies [23,26,27].

Hybrid watermilfoil has been documented in midwestern and western U.S. states [6,11,28] and the province of Ontario, Canada [29]. Hybrid watermilfoil appears most common in the Midwest [11]. Its occurrence has been documented in Minnesota since 2002 [3] and four lakes since 2007 [6], although we do not know how common or widespread infestations in Minnesota are. Hybrid watermilfoil is widespread in Michigan [30] and ~150 hybrid watermilfoil infestations have been identified in Wisconsin [20]. Although hybrid watermilfoil populations have long been documented, the spatial distribution in Minnesota and habitat characteristics associated with hybrid presence have yet to be investigated. Efforts to distinguish factors that promote the formation of hybrids and those that allow the persistence of hybrids [5] will be particularly insightful.

Previous studies in Minnesota have analyzed predictors of Eurasian watermilfoil invasions [31,32]. Roley and Newman [31] determined that Eurasian watermilfoil occurrences were most accurately predicted by distance to the nearest invaded lake and duration of that invasion. Their study also identified other characteristics including lake size, alkalinity, Secchi depth, and lake depth as significant predictors of Eurasian watermilfoil occurrence [31]. Confirmed Eurasian watermilfoil infestations in Minnesota have also been determined to be confounded by human population densities as well as associated with interstate highways [32]. Factors influencing the occurrence of hybrid watermilfoil have not been assessed; therefore we are unsure whether hybrid exhibits similar presence as Eurasian, or is very different from its exotic or native parents. Thum et al. [30] assessed within and among lake and region genetic diversity in Eurasian and hybrid watermilfoil in Michigan and Minnesota but did not assess the factors influencing the diversity or occurrence of these taxa.

The objective of our study was to describe the geographic distribution of hybrid watermilfoil among Minnesota lakes and relate this to environmental and infestation-associated variables. We aim to determine if hybrid watermilfoil is geographically widespread across the state or more likely to be present in the seven county Twin Cities Metro area, hereafter referred to as the "Metro area" (Appendix A Table A1). We also assess if it is more likely to occur in lakes with native northern versus Eurasian watermilfoil and determine the influence of age of infestation on hybrid watermilfoil presence. Lastly, we determine if human interaction such as boat access and herbicidal management are significantly associated with the presence of hybrid watermilfoil. By conducting these analyses to assess factors associated with confirmed hybrid watermilfoil invasions we can better inform prevention and control efforts and gain some insight on hybridity as it relates to invasive spread and establishment.

## 2. Materials and Methods

### 2.1. Study Sites

To determine the distribution of hybrid (and coincidentally Eurasian and northern) watermilfoil in Minnesota we sampled 62 lakes with varying duration of infestation and size in 24 counties across the state. We determined the number of lakes to sample per county based on the relative numbers of lakes with documented Eurasian watermilfoil infestations (including hybrid) as of 2017 from the Minnesota Department of Natural Resources' (MNDNR) infested waters list: https://www.dnr.state.mn.us/invasives/ais/infested.html (Appendix A Table A1). This method of lake selection ensured our survey represented the statewide distribution of Eurasian watermilfoil infestations, but limited surveys in counties where there were relatively few infestations. As a result, some counties fell outside of our sampling regime. The slight overrepresentation in Hennepin county is due to the inclusion of several bays of Lake Minnetonka.

Upon determining the number of lakes per county to sample, we selected (stratified) lakes based on size, maximum depth, and durations of infestation. Lakes sampled ranged from 12.5 to 51,891 ha in size, 2.5 to 135 m in maximum depth, and the durations of infestation ranged from 1 to 31 years (Appendix A Table A2). Because the MNDNR does not differentiate between Eurasian and hybrid when identifying invasive watermilfoil infestations, the year first infested may be based on either Eurasian or hybrid. We also sampled and recorded the presence of northern watermilfoil at each location, but our data do not fully reflect the distribution of northern watermilfoil in Minnesota because we sampled from only lakes listed as Eurasian/hybrid infested.

### 2.2. Field Sampling and Data Collection

At each lake we navigated to ~100 pre-selected random points distributed within a predefined littoral zone (depth ≤4.6 m). At each survey point, taxa were identified visually based on morphological features and leaflet counts. The following leaflet counts were used to identify each taxon: Eurasian 14–21 leaflet pairs, northern 5–9 pairs, and hybrid 10–13 pairs [6]. Plants were identified visually in order to collect representative samples for each unique watermilfoil taxon at each survey point. For example, if at a point we found a plant that looked like northern and another that looked like hybrid watermilfoil, a stem was collected for each plant. Stems were then placed in a labeled sealable bag on ice in a cooler. At each surveyed point the depth and number of plant stems per taxon collected were recorded. It is important to note that if we did not detect a particular taxon at a lake, this does not mean it was not present. We sampled thoroughly within the littoral zone, but it is possible that we did not identify all watermilfoil taxa present at surveyed lakes. Lakes may have contained a particular taxon of watermilfoil, but the abundance could have been below our detection limit and not found during our surveys.

### 2.3. Genetics

Total genomic DNA was extracted from a subset of collected plant samples using DNeasy Plant Mini Kits (Qiagen). When 20 or fewer plants were collected from a lake, all the samples collected were analyzed. We randomly subsampled at least 20 plants for analysis from lakes with more samples. To distinguish Eurasian, hybrid, and northern watermilfoil, plants were identified to taxon using a genetic assay based on internal transcribed spacer (ITS) DNA sequence [9,33]. Genetic variation was quantified for sampled plants and specific clones were delineated using eight microsatellite markers developed by Wu et al. [34] (Myrsp 1, Myrsp 5, Myrsp 9, Myrsp 12, Myrsp 13, Myrsp 14, Myrsp 15, and Myrsp 16). The protocols in Wu et al. [33] were used to amplify each microsatellite locus. Fragment analysis was completed on fluorescently labeled microsatellite polymerase chain reaction (PCR) products by the University of Illinois Urbana-Champaign Core Sequencing Facility using an ABI 3730xl sequencer. GeneMapper, version 5.0 (Applied Biosystems, Waltham, MA, USA), was used to score microsatellites. Microsatellites were used as dominant, binary data (i.e., presence or absence

of each possible allele at each locus) using the R-package POLYSAT [35]. Distinct genotypes were delineated using Lynch distances and a threshold of zero in POLYSAT [35].

### 2.4. Data Analysis

Based on genetically determined taxon identifications, all surveyed lakes were mapped with ArcGIS 10.5 to indicate presence/absence of each watermilfoil taxon. The geographic distribution of all collected watermilfoil was determined, as well as relative distance to nearest infestation. The distance to nearest infestation was determined by calculating the distance between our surveyed lake to the nearest Eurasian infestation (based on the 2017 MNDNR infested waters list). Infestations were assessed to determine relative occurrence in the Metro area (Appendix A Table A1) versus Greater Minnesota (all counties outside the Metro area). Statewide co-occurrence patterns were compared using a chi-squared test. Watermilfoil genotype richness was calculated for each lake using rarefaction. The rarefaction score was calculated using the rarefy function of the VEGAN package in R. This method calculates an expected species richness based on sample size.

To determine the influence of environmental and infestation-associated variables with the presence of hybrid watermilfoil in Minnesota and to make comparisons between lakes, the following factors were assessed for each lake (or bay of Lake Minnetonka): age of infestation, number of vehicle/trailer parking spaces at water accesses, lake area, maximum depth and littoral area (water depth ≤4.6 m) as obtained from the MNDNR's LakeFinder database (https://www.dnr.state.mn.us/lakefind/index.html). Secchi depth and trophic state index data were obtained from the Minnesota Pollution Control Agency (MPCA) lake and stream water quality assessment database (https://cf.pca.state.mn.us/water/watershedweb/wdip/index.cfm). Data were based on the 10-year average from state index data collected between June and September 2008 to 2017. Lakes were given watermilfoil management ratings on a scale of zero to three to describe the extent of watermilfoil management, which included only herbicidal control, based on MNDNR permit approval data from 2009 to 2017. A zero indicates no management during this period, one indicates spot treatments (less than 2.5% of the littoral area), two indicates intermediate management (2.5%–10% of littoral area) and three indicates lake wide treatments (greater than 10% of littoral area) targeting watermilfoil.

A total of six lakes were excluded from analyses of these lake attributes; two lakes contained no watermilfoil and in four lakes sampling efforts were not broadly or randomly distributed. However all lakes where we found watermilfoil were included in the distribution analysis to indicate presence/absence. To calculate average values for the analyzed variables, lakes were grouped based on the presence of each watermilfoil taxon. For example, EWM lakes include all lakes where Eurasian watermilfoil was found, HWM lakes includes all lakes where hybrid watermilfoil was found and NWM lakes include all lakes where northern watermilfoil was found. This categorization makes it possible for the same lake to be present in more than one group, if multiple watermilfoil taxa were found at a lake. The averages were calculated for all analyzed variables at the statewide level, as well as by separating lakes by region (Metro area and Greater Minnesota).

To ensure that the number of parameters reasonably align with our sample size, variables were separated into two groups for analysis: environmental (lake area, maximum depth, Secchi depth, littoral area, and trophic state index) and infestation-associated variables (age of infestation, number of parking spaces at water access, management score, and distance to nearest infestation). Environmental variables were identified as physical lake characteristics and infestation-associated variables were identified
assessed using a two-way multivariate analysis of variance (MANOVA) to determine if significant differences existed across taxon, region, and the interaction of taxon by region. A Tukey's honest significant difference (Tukey's HSD) test was then used to assess significant relationships between variables. The lack of normality seen in the distribution of some of our dependent variables was a result of skewness, rather than outliers. Therefore, the use of a MANOVA for our analysis is appropriate because the overall *F*-test is robust to skewness [36].

Logistic regression analysis (LRA) was used to identify variables associated with the presence and absence of each watermilfoil taxon across surveyed lakes. This analysis was used to model the probability of the presence and absence of each watermilfoil taxon in relation to the assessed environmental and infestation-associated variables. The LRA was performed with the software package R version 3.6.1 using the 'glm' function. The results of LRA indicate which variables are associated with the increased probability of the presence of each watermilfoil taxon. A *p*-value of 0.10 was used to determine significance for all assessments. We present *p*-values for all statistics so readers can make their own interpretation.

## 3. Results

### 3.1. Watermilfoil Distribution

Of the total 62 lakes sampled, 43 contained Eurasian, 28 contained hybrid, 21 contained northern, and no watermilfoil was found in two lakes. Eurasian watermilfoil was evenly distributed across the state (Figure 1). Hybrid watermilfoil was most commonly found in lakes in the Metro area (82%). Of the 28 lakes that we found containing hybrid watermilfoil, 13 had only hybrid and no other watermilfoil taxa, and the remaining 15 had some combination with either Eurasian, northern, or both (Table 1). Northern watermilfoil was most commonly found in lakes in Greater Minnesota (68%).

We found various taxa combinations in lakes where watermilfoil was found (Table 1). Eurasian watermilfoil was commonly found with either northern (60%) or hybrid (40%). There was no significant difference in Eurasian watermilfoil co-occurrence patterns ($X^2$, $p > 0.1$). Hybrid watermilfoil-only infestations were mostly present in the Metro area (91%); only one hybrid exclusive infestation was found in Greater Minnesota. Hybrid watermilfoil was most commonly found in lakes with Eurasian (12 lakes) compared to northern (7 lakes). Hybrid watermilfoil was more commonly found in lakes with Eurasian ($X^2$, $p = 0.08$). Northern watermilfoil was also more commonly found with Eurasian ($X^2$, $p = 0.002$). We found four lakes that contained all three taxa, half of which were in the Metro area and half in Greater Minnesota (Table 1).

**Table 1.** Number of lakes with various combinations of watermilfoil taxa identified (EWM = Eurasian watermilfoil, HWM = hybrid watermilfoil, NWM = northern watermilfoil). Lake counts are grouped based on region (Greater Minnesota and Metro area).

| | EWM Only | HWM Only | NWM Only | EWM and HWM | NWM and HWM | EWM and NWM | All Three Taxa | Total |
|---|---|---|---|---|---|---|---|---|
| Greater Minnesota | 8 | 1 | 1 | 0 | 2 | 10 | 2 | 24 |
| Metro area | 10 | 12 | 0 | 8 | 1 | 3 | 2 | 36 |
| Total | 18 | 13 | 1 | 8 | 3 | 13 | 4 | 60 |

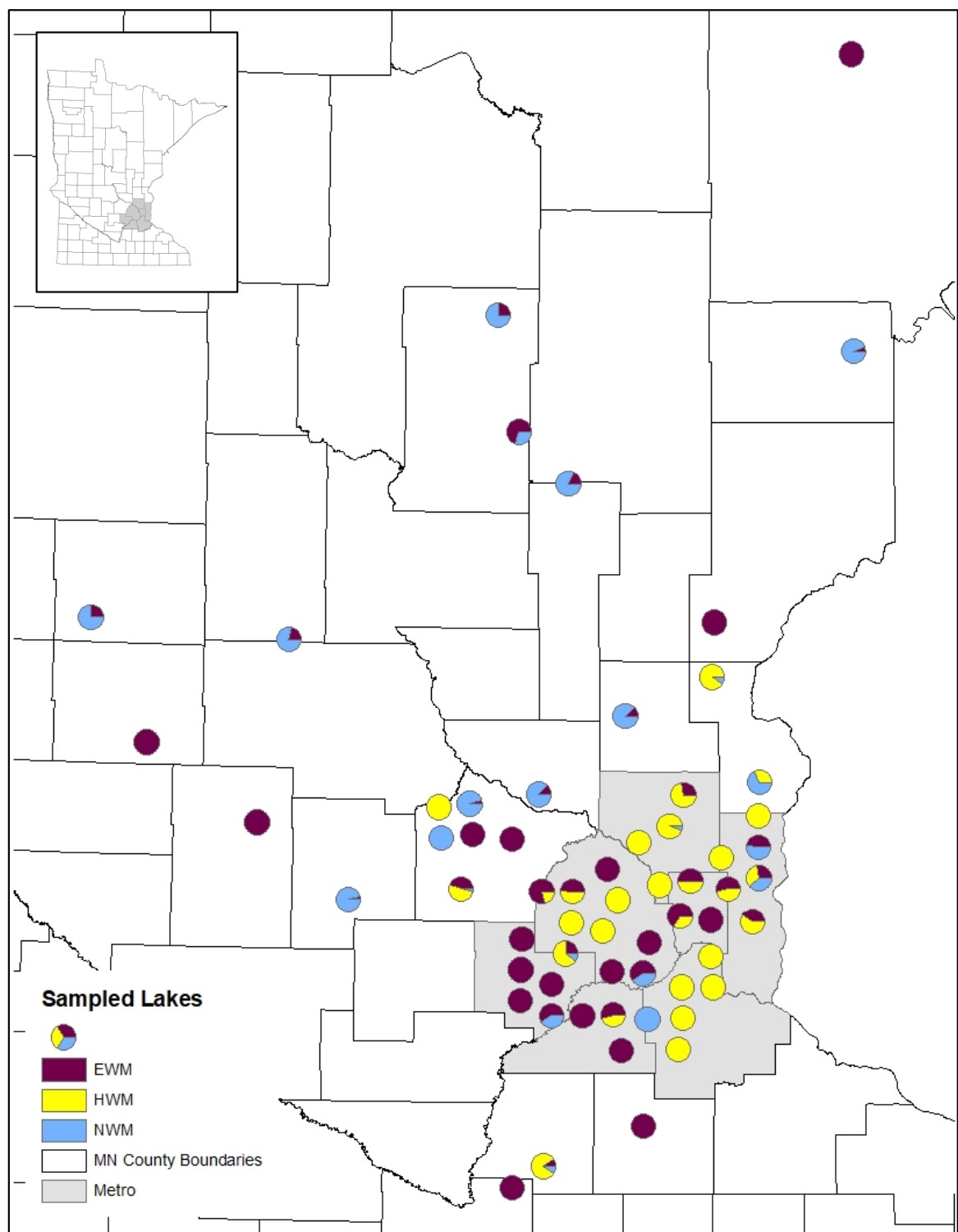

**Figure 1.** Statewide occurrence of Eurasian (EWM, purple), hybrid (HWM, yellow), and northern (NWM, blue) watermilfoil proportions in sampled lakes. Metro area counties are shaded in gray.

*3.2. Watermilfoil Genotyping*

Based on the number of unique genotypes identified, Eurasian watermilfoil was the least diverse (6 genotypes), hybrid was intermediate (51 genotypes) and northern was the most diverse (81 genotypes; Table 2). Based on the rarefaction analysis, northern watermilfoil populations were significantly more diverse than Eurasian and hybrid watermilfoil ($p = 0.01$). Lakes where we found northern typically had multiple northern genotypes (86% of lakes with 2 to 8 genotypes) whereas 93% of lakes with Eurasian had only one Eurasian watermilfoil genotype. Lakes with hybrid watermilfoil typically had a single hybrid genotype (57%), although we found 43% of lakes had two or more hybrid genotypes (Appendix A Table A3).

A total of 18 unique hybrid watermilfoil genotypes were found in the three bays of Lake Minnetonka (Grays, North Arm, and Smiths). We found three hybrid watermilfoil genotypes that were repeated among bays. Despite common hybrid watermilfoil genotypes, North Arm and Grays Bay each had four unique hybrid genotypes, and Smiths Bay had seven. The diversity in the Minnetonka bays accounted for 35% of the hybrid watermilfoil genotypes we identified.

**Table 2.** Number of genotypes found for each taxon from assessed watermilfoil samples. Average watermilfoil rarefied genotype richness per lake and standard error (EWM = Eurasian watermilfoil, HWM = hybrid watermilfoil, NWM = northern watermilfoil).

| Taxon | Total Number of Unique Genotypes | Average Watermilfoil Genotype Richness/Lake |
|-------|----------------------------------|---------------------------------------------|
| EWM | 6 | 2.5 ± 0.3 |
| HWM | 51 | 2.7 ± 0.3 |
| NWM | 81 | 3.5 ± 0.3 |

Most lakes containing Eurasian watermilfoil had the same genotype. Hybrid genotypes were not typically repeated among lakes, although we found four hybrid watermilfoil genotypes that were in more than one lake. Lakes Elmo and Coon shared a hybrid watermilfoil genotype and Lac Lavon shared a hybrid watermilfoil genotype with Cobblestone (Figure 2). Lac Lavon also shared a different hybrid watermilfoil genotype with Alimagnet (Figure 2). We found one hybrid watermilfoil genotype in seven different lakes: Bald Eagle, Bone, Fish, Josephine, Otter, South Lindstrom, and White Bear (Figure 2). No northern watermilfoil genotypes were found in more than one lake. Although Minnetonka bays had many hybrid genotypes and shared three genotypes, no Minnetonka hybrid genotypes were found in other lakes.

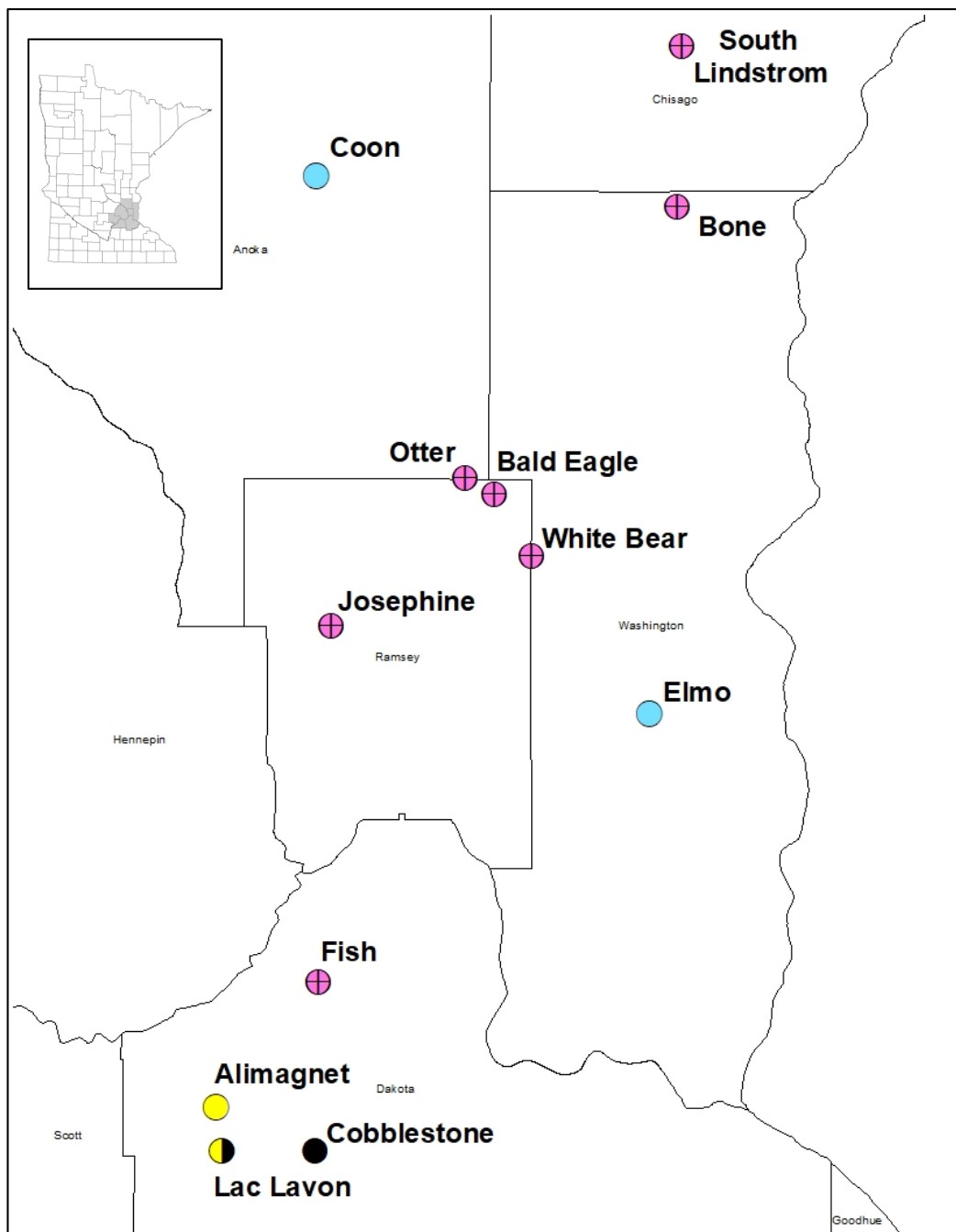

**Figure 2.** Distribution of shared hybrid watermilfoil genotypes in sampled lakes.

*3.3. Environmental Variables Analysis*

The environmental variables MANOVA indicated that there were no significant differences across lakes based on taxon presence, region, or the interaction between taxon and region (Table 3). The LRA indicated that northern watermilfoil presence was positively associated with littoral area ($p = 0.04$; Table 4) and northern was more likely to be found in lakes with shallower maximum depth ($p = 0.07$) and smaller lake area ($p = 0.09$). No environmental variables were found to be associated with Eurasian and hybrid watermilfoil presence (Table 4).

**Table 3.** Mean values and standard errors for environmental variables of lakes classified as containing either Eurasian (EWM), hybrid (HWM), or northern (NWM) watermilfoil. Multivariate analysis of variance (MANOVA) *p*-values are given for taxon, region, and taxon by region.

| | | Number of Lakes | Lake Area (ha) | Max Depth (m) | Secchi Depth (m) | Littoral Area (ha) | Trophic State Index |
|---|---|---|---|---|---|---|---|
| **EWM** | Statewide | 41 | 299 ± 62 | 17.5 ± 3.2 | 2.5 ± 0.3 | 159 ± 28 | 53 ± 2 |
| | Metro area | 21 | 231 ± 55 | 15.2 ± 2.1 | 2.4 ± 0.3 | 142 ± 35 | 54 ± 2 |
| | Greater Minnesota | 20 | 379 ± 116 | 20.4 ± 6.6 | 2.7 ± 0.4 | 178 ± 45 | 52 ± 2 |
| **HWM** | Statewide | 26 | 202 ± 45 | 12.3 ± 1.5 | 2.4 ± 0.2 | 122 ± 29 | 53 ± 1 |
| | Metro area | 21 | 164 ± 33 | 12.7 ± 1.9 | 2.5 ± 0.2 | 109 ± 32 | 52 ± 2 |
| | Greater Minnesota | 5 | 363 ± 186 | 10.9 ± 1.4 | 1.8 ± 0.4 | 174 ± 68 | 60 ± 1 |
| **NWM** | Statewide | 21 | 314 ± 52 | 14.3 ± 1.7 | 2.8 ± 0.3 | 177 ± 31 | 51 ± 2 |
| | Metro area | 6 | 261 ± 109 | 12.9 ± 3.3 | 3.0 ± 0.7 | 175 ± 71 | 49 ± 3 |
| | Greater Minnesota | 15 | 321 ± 60 | 15.6 ± 2.1 | 2.6 ± 0.3 | 167 ± 34 | 51 ± 2 |
| **MANOVA** | | | | | | | |
| *p*-value | Taxon | | 0.519 | 0.399 | 0.566 | 0.649 | 0.562 |
| | Region | | 0.134 | 0.420 | 0.765 | 0.409 | 0.398 |
| | Taxon by Region | | 0.908 | 0.755 | 0.568 | 0.898 | 0.229 |

Note: Lake types include all lakes with the taxon present and therefore a lake may be represented in more than one category.

**Table 4.** Logistic regression analysis (LRA) assessment of Eurasian (EWM), hybrid (HWM), and northern watermilfoil (NWM) presence-absence based on environmental variables. Output includes the model intercept and coefficient and *p*-value associated with each variable.

| | Coefficient | *p*-Value |
|---|---|---|
| **EWM** | | |
| <u>**Intercept**</u> | −0.75 | 0.90 |
| Lake area | 0.02 | 0.12 |
| Littoral area | −0.02 | 0.24 |
| Maximum depth | 0.05 | 0.36 |
| Secchi depth | −0.23 | 0.71 |
| Trophic state index | 0.02 | 0.84 |
| **HWM** | | |
| <u>**Intercept**</u> | 0.62 | 0.90 |
| Lake area | −0.004 | 0.32 |
| Littoral area | 0.007 | 0.32 |
| Maximum depth | −0.04 | 0.30 |
| Secchi depth | 0.18 | 0.72 |
| Trophic state index | −0.01 | 0.85 |
| **NWM** | | |
| <u>**Intercept**</u> | −2.72 | 0.61 |
| Lake area | −0.005 | 0.09 * |
| Littoral area | 0.01 | 0.04 * |
| Maximum depth | −0.05 | 0.07 * |
| Secchi depth | 0.92 | 0.11 |
| Trophic state index | 0.002 | 0.976 |

* significant (*p* < 0.10).

### 3.4. Infestation-Associated VariablesAanalysis

The MANOVA of infestation-associated variables indicated there were significant differences across region for age (*p* = 0.004), parking (*p* = 0.006), and distance to nearest infestation (*p* = 0.018; Table 5). Infestations in the Metro area were significantly older, and the lakes had more parking spaces at the water access, and were closer to other documented infestations (Tukey's HSD). There were no significant differences among lakes for all infestation-associated variables based on taxon presence or the interaction of region and taxon (MANOVA: Table 5).

The LRA revealed that the presence of hybrid ($p = 0.01$) and northern ($p = 0.001$) watermilfoil was associated with region (Table 6). Hybrid watermilfoil presence was positively associated with lakes located in the Metro area (Tables 5 and 6). Northern watermilfoil presence was positively associated with lakes in Greater Minnesota (Tables 5 and 6). The LRA of hybrid watermilfoil indicated that age of infestation ($p = 0.73$), parking spaces ($p = 0.44$), and management scores ($p = 0.34$) were not significantly associated with hybrid presence. Although hybrid watermilfoil presence had greater probability in the Metro area, this did not directly translate to hybrid infestations being significantly closer to other invasive watermilfoil infestations ($p = 0.43$). Eurasian watermilfoil presence was not significantly associated with any infestation-associated variables according to the LRA (Table 6).

Because the multivariable LRA could mask important factors that may be individually important, infestation-associated variables were also analyzed individually using the LRA. Based on this analysis we found that hybrid watermilfoil presence was significantly associated with parking ($p = 0.09$) and age ($p = 0.07$). Lakes containing hybrids were found to have more parking spaces at the water access and be older infestations. However, after the inclusion of region in this analysis, parking and age were no longer significant. This indicates that age and parking are confounded with region and suggests region is the overarching driver.

**Table 5.** Mean values and standard errors for infestation-associated variables of lakes classified as containing either Eurasian (EWM), hybrid (HWM), or northern (NWM) watermilfoil and *p*-values for taxon, region, and taxon by region.

| | | Number of Lakes | Age of Infestation (Years) | Parking Spaces at Water Access | Management Score | Distance to Nearest Infestation (km) |
|---|---|---|---|---|---|---|
| **EWM** | Statewide | 41 | 16.6 ± 1.3 | 22.0 ± 4.1 | 1.0 ± 0.1 | 20.8 ± 3.5 |
| | Metro area | 21 | 19.7 ± 1.8 | 31.9 ± 7.1 | 0.9 ± 0.2 | 7.9 ± 1.3 |
| | Greater Minnesota | 20 | 13.2 ± 1.7 | 11.5 ± 2.6 | 1.0 ± 0.2 | 34.1 ± 5.6 |
| **HWM** | Statewide | 26 | 19.2 ± 1.8 | 27.7 ± 6.4 | 1.2 ± 0.2 | 11.3 ± 2.2 |
| | Metro area | 21 | 20.2 ± 2.1 | 29.5 ± 7.5 | 1.3 ± 0.2 | 7.2 ± 0.8 |
| | Greater Minnesota | 5 | 15.0 ± 2.9 | 21.6 ± 10.9 | 1.2 ± 0.6 | 30.7 ± 7.1 |
| **NWM** | Statewide | 21 | 17.8 ± 1.9 | 23.0 ± 4.4 | 1.1 ± 0.2 | 29.4 ± 5.3 |
| | Metro area | 6 | 21.2 ± 4.1 | 35.8 ± 8.5 | 1.0 ± 0.4 | 12.9 ± 3.0 |
| | Greater Minnesota | 15 | 16.4 ± 2.1 | 17.8 ± 4.6 | 1.1 ± 0.3 | 36.0 ± 6.6 |
| **MANOVA *p*-value** | | | | | | |
| Taxon | | | 0.341 | 0.671 | 0.578 | 0.338 |
| Region | | | 0.004 * | 0.006 * | 0.354 | 0.018 * |
| Taxon by Region | | | 0.897 | 0.703 | 0.205 | 0.772 |

Note: Lake types include all lakes with the taxon present and therefore a lake may be represented in more than one category. * significant ($p < 0.10$).

**Table 6.** Logistic regression analysis (LRA) assessment of Eurasian (EWM), hybrid (HWM), and northern watermilfoil (NWM) presence-absence based on infestation-associated variables. Output includes the model intercept and coefficient and *p*-value associated with each variable.

| | Coefficient | *p*-Value |
|---|---|---|
| **EWM** | | |
| **Intercept** | 0.43 | 0.65 |
| Region | 1.21 | 0.11 |
| Age of infestation | 0.01 | 0.71 |
| Parking spaces | 0.01 | 0.45 |
| Management score | −0.51 | 0.13 |
| Distance to nearest infestation | 0.06 | 0.55 |

**Table 6.** *Cont.*

|  | Coefficient | *p*-Value |
|---|---|---|
| **HWM** | | |
| **Intercept** | 0.04 | 0.97 |
| Region | −1.71 | 0.01 * |
| Age of infestation | 0.01 | 0.73 |
| Parking spaces | 0.01 | 0.44 |
| Management score | 0.31 | 0.34 |
| Distance to nearest infestation | −0.07 | 0.43 |
| **NWM** | | |
| **Intercept** | −2.10 | 0.04 * |
| Region | 2.48 | 0.001 * |
| Age of infestation | 0.01 | 0.78 |
| Parking spaces | 0.02 | 0.19 |
| Management score | −0.08 | 0.81 |
| Distance to nearest infestation | −0.02 | 0.63 |

* significant ($p < 0.10$).

## 4. Discussion

Hybrid watermilfoil found in a lake can be the result of in situ hybridization between the native northern and invasive Eurasian watermilfoil (i.e., formation [5]), or clonal spread from elsewhere. However, for hybrids to be found and to spread, they must persist, whether developed in situ or introduced from elsewhere. Once formed, populations can persist through either sexual or vegetative reproduction and potentially spread to new lakes, although they could go extinct or below detection limits. Invasive Eurasian and native northern watermilfoil populations can also be clonally spread or sexually reproduce, although as with hybrids, their persistence will depend on suitable habitat and lack of displacement. This framework of formation and persistence [5] with dispersal by clonal spread will inform our discussion of factors influencing the occurrence and distribution of Eurasian, hybrid and northern watermilfoil. We assume that plants identified as the same microsatellite genotype share ancestry via asexual reproduction (i.e., are clones, or the same genets), and that different microsatellite genotypes are the result of sexual reproduction. Although there is some error in determining microsatellite genotypes [37], it is unlikely that two individuals with the same microsatellite genotype share ancestry via sexual reproduction but are identical for microsatellite markers by chance [37]. Furthermore, it is highly unlikely that multiple occurrences of the same microsatellite genotype would all be separate genotypes that are identical by chance. Conversely, populations composed of unique genotypes were likely developed as the result of sexual reproduction [38], however there is a possibility of clonal introduction from elsewhere that we have not identified.

Hybrid watermilfoil was found in about half of the lakes surveyed and was concentrated in the Metro area, but we also found hybrids in Greater Minnesota. Eurasian was found in most lakes surveyed (70%) and widely distributed across infested lakes in the state, whereas northern watermilfoil was found in a third of lakes and was most common in Greater Minnesota. The types of lakes that hybrid watermilfoil inhabited were very similar to those with Eurasian and northern in regards to our analyzed lake attributes; lake attributes do not appear to determine differential persistence of hybrids. Wu et al. [39] quantified and compared the climate niches of Eurasian and northern watermilfoil in two co-occurring regions in their native range. They determined that hybrid watermilfoil was more commonly found in the region where Eurasian and northern watermilfoil occupied similar environments, because of the increased likelihood of contact between the two taxa [39]. Hybridization was less likely to occur where Eurasian and northern watermilfoil occupied areas unique to their respective niche. We suspect most lakes in Minnesota with Eurasian previously supported northern watermilfoil providing an opportunity to develop hybrid watermilfoil populations in Minnesota.

We did not find evidence that hybrids have a separate or special niche compared to Eurasian and northern watermilfoil.

Rather than be associated with environmental factors, the concentration of hybrid watermilfoil in the Metro area may be explained by the process of hybrid formation as a result of sexual reproduction between Eurasian and northern watermilfoil. This requires that both taxa are established within a lake. The hybrid watermilfoil present can persist and diversify as a result of sexual reproduction between hybrids [40]. These processes may take more time, thus preventing hybrid watermilfoil from having more widespread distribution [7]. Based on this rationale we initially predicted that hybrid watermilfoil would more likely be present in lakes with older periods of infestation, but our analysis did not find age to be significant. Hybrid watermilfoil presence was not associated with older invasions when region was included.

Age may have not been associated with hybrid watermilfoil presence due to clonal spread that commonly occurred in the Metro area. We found common hybrid watermilfoil genotypes were present in multiple lakes/bays only within the Metro area. We did not see this in lakes of Greater Minnesota where we found hybrid watermilfoil. Clonal spread of hybrid watermilfoil more likely results in newer (younger) infestations as opposed to in situ formed hybrids, which may be indicative of older infestations. Age was a significant predictor of hybrid watermilfoil presence when considered alone, but not once region was included in the analysis. Thus, age may be a factor associated with hybrid watermilfoil presence, but our analysis could not detect this relationship in part due to few Greater Minnesota occurrences. In order to determine if age is significantly associated with hybrid watermilfoil occurrences, age should be assessed separately for presumed clonal and in situ populations.

Hybrid watermilfoil presence was not associated with closer infestations. Rather, hybrid watermilfoil presence was associated with Metro area lakes specifically. This suggests that human mediated interactions in the Metro area are linked to hybrid watermilfoil spread and/or formation. Lakes in the Metro area are more likely to be infested with hybrid watermilfoil, increasing the likelihood of spreading it from one lake to another. These findings are consistent with those of Guo [41] who indicated that anthropogenic interactions can stimulate the formation and persistence of hybrid plants. The Metro area is more densely populated than Greater Minnesota, therefore lakes tend to be closer to interstate highways and major cities. Hybrid watermilfoil was more commonly found in the Metro area, which may be associated with human population densities, similar to previous findings regarding Eurasian infestations [32].

We predicted, as suggested by LaRue et al. [40], that hybrid watermilfoil presence would be associated with intensive watermilfoil herbicidal management histories. Repeated herbicide treatments may select for herbicide tolerant hybrids, if present, and therefore these watermilfoil populations are more likely to be dominated by hybrids. However, management score was not significantly associated with hybrid watermilfoil presence. This suggests that the hybrid watermilfoil populations in Minnesota have not directly developed in response to management pressures. It is important to note that hybrid watermilfoil encompasses many different genotypes and they are not equal in terms of herbicide response [42]. Herbicide-resistant hybrid watermilfoil genotypes have been identified in Michigan [23,24,26] but not yet in Minnesota. Therefore, lakes should be managed alongside active genetic characterization to assess population changes that arise. Frequent monitoring of hybrid watermilfoil populations is important to verify the efficacy of watermilfoil management and detect tolerant genotypes.

Eurasian watermilfoil occurrences were widespread in Minnesota, which aligns with the fact that we surveyed lakes categorized as Eurasian-infested. Perhaps not surprisingly, no environmental or infestation-associated variables explained Eurasian watermilfoil occurrence. Northern watermilfoil most commonly occurred in Greater Minnesota and its presence was associated with several lake attributes. The unique watermilfoil genotypes are the result of sexual reproduction between plants, whereas, microsatellite-identified identical genotypes are most likely the result of clonal spread or asexual reproduction. The northern watermilfoil populations we identified were more diverse

than Eurasian, indicating sexual reproduction is common within northern populations [see also 38]. This contrasts with Eurasian watermilfoil infestations, which we found to be due to clonal spread and persistence. This is consistent with previous studies that have found predominantly clonal spread and lesser importance of sexual reproduction and persistence for Eurasian watermilfoil [11,38]. In contrast, we found no evidence of clonal spread of northern watermilfoil and populations were persisting with sexual reproduction generating some diversity and persistence as also noted by [38].

Thum et al. [30] found more genetic diversity in Eurasian populations in Michigan (18 genotypes). Eurasian watermilfoil populations have been present in Michigan since the early 1960s [43] whereas they were first noted in the late 1980s in Minnesota. There may not have been enough time for sexually reproduced genotypes of Eurasian to develop, persist and spread in Minnesota. The lack of diversity in our Eurasian watermilfoil populations could also be an issue of sexually produced Eurasian plants being incapable of persisting within our lakes. The one widespread Eurasian genotype (found in Michigan and Minnesota) may have reduced viability from selfing, but this clone may possess the ability to be successful over a broad range of environmental conditions and, therefore, clonal persistence and spread is dominant [44].

More than half of the hybrid watermilfoil occurrences we identified were unique occurrences. This could be due to within-lake hybrid formation or to colonization by a genet produced elsewhere whose source was not identified. Genetically distinct hybrid watermilfoil genotypes could be produced as a result of hybridization of Eurasian and northern, sexual reproduction between hybrids, or introgression in which hybrids backcross with either Eurasian or northern [40]. The genetic analysis methods we used are not able to identify whether or not the hybrid watermilfoil we collected are the $F_1$ generation. Therefore, it is unclear whether hybrids are sexually reproducing more often or if Eurasian and northern watermilfoil tend to hybridize if both are present within a lake.

Hybrid watermilfoil occurrence can also be due to clonal spread, which we found occurred in at least 12 lakes. We hypothesized that this may be related to boater traffic, although we did not find that public lake access was significantly associated with hybrid watermilfoil presence. Six of the 12 lakes with hybrid spread were adjacent or within the same drainage, but other lakes were further apart and perhaps on some circuit of anglers or lake professionals. Clonal spread may also be caused by transport by waterfowl [45].

Hybrid watermilfoil-only infestations occur quite commonly, especially in the Metro area. This contrasts the findings of Sturtevant et al. [28] who found that only two of 15 lakes surveyed in MI and IN contained only hybrid watermilfoil and no other watermilfoil taxon. This could be the result of our increased sampling efforts (62 vs. 15 lakes) in comparison to Sturtevant et al. [28]. Thum et al. [30] found hybrid alone in 15 of the 41 lakes they sampled in Michigan. The hybrid-only infestations we identified may have been produced sexually by Eurasian and northern watermilfoil and over time hybrids may have outcompeted their parental taxa. Another explanation may be that these hybrid watermilfoil introductions are the result of hybrid clonal spread, and our survey did not locate the clonal source, but it seems unlikely that we would not detect that many hybrid genotypes in other lakes.

Eurasian and hybrid watermilfoil commonly co-occurred in our lakes, as they did in the Michigan and Indiana lakes studied by Sturtevant et al. [28]. In contrast Moody and Les [6] did not find the co-occurrence of Eurasian and hybrid watermilfoil in Minnesota, although they did in two Wisconsin lakes and one in Idaho. Very few (seven) lakes in our survey contained both northern and hybrid watermilfoil, partly as a result of their occurrences being more common in different regions of the state. Hybrid watermilfoil may be less commonly occurring with northern because northern may be outcompeted by invasive Eurasian (or hybrid) over time [46] or because northern may be eliminated by repeated herbicide treatments to control invasive watermilfoil. Reduced presence of northern watermilfoil has been found in previous studies [6,28] and warrants further investigation as the spread of hybrids increases. Northern watermilfoil likely had a greater presence in the Metro area than it

currently does, but populations are decreasing through either competition, genomic contamination [6], or the use of herbicides [47].

All three watermilfoil taxa were present in four lakes (Bald Eagle, German, Howard, and Smith's Bay of Lake Minnetonka). In two lakes (German and Howard) taxa primarily consisted of Eurasian and hybrid watermilfoil and only a single northern sample was collected. In contrast, Bald Eagle and Smith's Bay had relatively similar abundances of all three watermilfoil taxa. Co-occurrence of watermilfoil taxa may be influenced by spatial distribution of taxa within the lake [28]. Lower abundances of northern watermilfoil in these lakes may be as a result of species overlap in which watermilfoil taxa inhabit common areas and are thereby directly impacted by competitive interactions for resources. This form of distribution increases the likelihood that northern watermilfoil abundance will decrease over time, because northern may be outcompeted by established invasive watermilfoil populations [48]. It is unclear what population dynamics exist in lakes with equal abundances of all three watermilfoil taxa. Segregation of watermilfoil taxa among areas when all three taxa co-occur within a lake may play a role in lessening the impact of competition between taxa. Analysis of within-lake composition and distribution of watermilfoil taxa is needed to better clarify these relationships. Our current genetic analyses approaches do not allow us to determine if the hybrids present in the lakes are the result of in situ reproduction by the parent taxa found to be present and the development of genomics approaches (e.g., [37]) may enhance our ability to do so.

## 5. Conclusions

Our survey extends the current knowledge of hybrid watermilfoil occurrence and spread with 28 confirmed occurrences in Minnesota. Hybrid watermilfoil is common and widespread in Minnesota, but is largely concentrated in the Metro area. Eurasian watermilfoil was widespread and equally distributed in the Metro area and Greater Minnesota. Eurasian watermilfoil infestations were primarily the result of clonal spread, whereas there was no evidence of clonal spread in native northern watermilfoil populations, which were more genetically diverse. We identified both genetically identical and unique hybrid watermilfoil genotypes, indicating that hybrid occurrences result from both clonal spread and in situ sexual reproduction. Hybrid watermilfoil occurrences were not found to be associated with a unique niche in comparison to Eurasian and northern watermilfoil; all three taxa inhabit similar environments. Furthermore, hybrids in our lakes do not appear to be related to management activities such as herbicidal control. Further investigation into the distribution of common hybrid watermilfoil genotypes would provide a better understanding as to whether the spread and persistence of specific genotypes is related to watermilfoil management or other factors.

**Author Contributions:** Conceptualization, R.M.N. and R.A.T.; methodology, R.M.N., R.A.T., and J.A.E.; software, J.A.E. and R.A.T.; validation, J.A.E., R.M.N., and R.A.T.; formal analysis, J.A.E., R.M.N., and R.A.T.; investigation, J.A.E., R.M.N., and R.A.T.; resources, R.M.N. and R.A.T.; data curation, J.A.E. and R.A.T.; writing—original draft preparation, J.A.E. and R.M.N.; writing—review and editing, J.A.E., R.M.N., and R.A.T.; visualization, J.A.E.; supervision, R.M.N.; project administration, R.M.N.; funding acquisition, R.M.N. and R.A.T. All authors have read and agreed to the published version of the manuscript.

**Funding:** This research was funded by the Minnesota Environmental and Natural Resources Trust Fund as recommended by the Minnesota Aquatic Invasive Species Research Center (MAISRC) and the Legislative-Citizen Commission on Minnesota Resources (LCCMR). Additional funding and resources for this project were provided by the USDA National Institute of Food and Agriculture, Hatch grant MIN-41-081, Minnesota Aquatic Invasive Species Research Center, and the University of Minnesota, Diversity of Views and Experiences and the College of Food, Agriculture and Natural Sciences Diversity fellowships through the Water Resources Science Graduate Program.

**Acknowledgments:** The authors thank Thomas Ostendorf, Alex Franzen, Matthew Gilkay, Kyle Blazek, and Jacob Olsen for assistance with sampling and data entry, and Jeffrey Korff, Gregory Chorak, and Jeff Pashnick for genetic analysis. Survey site suggestions and herbicide management data were provided by Keegan Lund, Kylie Cattoor, April Londo, Wendy Crowell, Eric Katzenmeyer, Tim Plude, Jon Hansen, Christine Jurek, Allison Gamble, Donna Perleberg, Richard Rezanka, and Rick Walsh of the MNDNR and James Johnson, Patrick Selter, Steve McComas, Justin Valenty, Brian Vlach, and Eric Fieldseth. Statistical and spatial analysis advice was provided by John Fieberg and Paul Bolstad of the University of MN. Comments by two anonymous reviewers helped us improve the paper.

**Conflicts of Interest:** The authors declare no conflict of interest.

## Appendix A

**Table A1.** Eurasian watermilfoil recorded infested lake counts by county out of 87 total Minnesota counties as of 2017, proportion of statewide occurrences, and number of lakes we sampled.

| County | Number of Infested Lakes | Proportion | Number Sampled | Proportion Sampled |
|---|---|---|---|---|
| * Hennepin | 46 | 0.144 | 11 | 0.177 |
| Wright | 40 | 0.125 | 7 | 0.113 |
| * Carver | 27 | 0.084 | 5 | 0.081 |
| * Ramsey | 27 | 0.084 | 6 | 0.097 |
| * Dakota | 22 | 0.069 | 5 | 0.081 |
| * Washington | 22 | 0.069 | 4 | 0.065 |
| * Anoka | 13 | 0.041 | 4 | 0.065 |
| Chisago | 12 | 0.038 | 2 | 0.032 |
| Crow Wing | 11 | 0.034 | 2 | 0.032 |
| * Scott | 10 | 0.031 | 2 | 0.032 |
| Meeker | 9 | 0.028 | 1 | 0.016 |
| Le Sueur | 8 | 0.025 | 1 | 0.016 |
| Rice | 7 | 0.022 | 1 | 0.016 |
| Itasca | 6 | 0.019 | 0 | 0.000 |
| Stearns | 6 | 0.019 | 0 | 0.000 |
| Kandiyohi | 5 | 0.016 | 1 | 0.016 |
| Sherburne | 5 | 0.016 | 1 | 0.016 |
| Blue Earth | 4 | 0.013 | 1 | 0.016 |
| Cass | 4 | 0.013 | 0 | 0.000 |
| Douglas | 4 | 0.013 | 1 | 0.016 |
| Isanti | 4 | 0.013 | 1 | 0.016 |
| Pine | 4 | 0.013 | 1 | 0.016 |
| Pope | 4 | 0.013 | 1 | 0.016 |
| Carlton | 3 | 0.009 | 1 | 0.016 |
| St. Louis | 3 | 0.009 | 1 | 0.016 |
| Morrison | 2 | 0.006 | 0 | 0.000 |
| Todd | 2 | 0.006 | 1 | 0.016 |
| Waseca | 2 | 0.006 | 0 | 0.000 |
| Winona | 2 | 0.006 | 0 | 0.000 |
| Kanabec | 1 | 0.003 | 0 | 0.000 |
| McLeod | 1 | 0.003 | 0 | 0.000 |
| Mille Lacs | 1 | 0.003 | 1 | 0.016 |
| Olmsted | 1 | 0.003 | 0 | 0.000 |
| Polk | 1 | 0.003 | 0 | 0.000 |

* located in Metro area.

**Table A2.** Lakes sampled in 2017–2018 including lake area, maximum depth and year of Eurasian infestation for each lake sampled.

| Lake name | County | Lake ID | Lake Area (hectares) | Maximum Depth (m) | Year of Eurasian Infestation |
|---|---|---|---|---|---|
| **Alimagnet** | Dakota | 19-0021 | 41.9 | 3.5 | 2012 |
| **Auburn** | Carver | 10-004401 | 117.6 | 25.6 | 1989 |
| **Bald Eagle** | Ramsey | 62-0002 | 423.6 | 11.0 | 1989 |
| **Ballantyne** | Blue Earth | 07-0054 | 150.0 | 17.7 | 2012 |
| **Bay** | Crow Wing | 18-0034 | 938.8 | 22.6 | 1992 |
| **Big Marine** | Washington | 82-0052 | 728.1 | 18.9 | 2004 |
| **Bone** | Washington | 82-0054 | 89.6 | 9.1 | 2006 |
| **Cedar** | Hennepin | 27-0039 | 66.3 | 15.5 | 1990 |
| **Cedar** | Wright | 86-0227 | 319.8 | 32.9 | 2010 |

**Table A2.** *Cont.*

| Lake name | County | Lake ID | Lake Area (hectares) | Maximum Depth (m) | Year of Eurasian Infestation |
|---|---|---|---|---|---|
| Christmas | Hennepin | 27-0137 | 108.1 | 26.5 | 1992 |
| Chub | Carlton | 09-0008 | 126.8 | 8.5 | 2009 |
| Cobblestone | Dakota | 19-0456 | 14.1 | 5.5 | 2011 |
| Constance | Wright | 86-0051 | 70.7 | 7.0 | 2016 |
| Coon | Anoka | 02-0042 | 599.4 | 8.2 | 2003 |
| Crooked | Anoka | 02-0084 | 46.5 | 7.9 | 1990 |
| East Rush | Chisago | 13-006901 | 599.2 | 7.3 | 1992 |
| Elmo | Washington | 82-0106 | 103.9 | 42.7 | 2005 |
| Emily | Crow Wing | 18-0203 | 291.7 | 4.0 | 2014 |
| Fish | Dakota | 19-0057 | 12.4 | 10.2 | 2009 |
| Fox | Rice | 66-0029 | 126.1 | 14.3 | 2009 |
| German | Le Seuer | 40-0063 | 320.4 | 15.5 | 2002 |
| Gervais | Ramsey | 62-0007 | 95.1 | 12.5 | 1995 |
| Gilbert Pit | St. Louis | 69-1306 | 102.8 | 135.0 | 1999 |
| Gilchrist | Pope | 61-0072 | 136.0 | 7.3 | 1996 |
| Green | Kandiyohi | 34-0079 | 2250.3 | 33.5 | 2000 |
| Ham | Anoka | 02-0053 | 77.1 | 6.7 | 2013 |
| Harriet | Hennepin | 27-0016 | 138.1 | 26.5 | 1991 |
| Howard | Wright | 86-0199 | 301.5 | 11.9 | 2003 |
| Independence | Hennepin | 27-0176 | 342.7 | 17.7 | 1989 |
| Indian | Wright | 86-0223 | 56.4 | 9.5 | 2003 |
| Josephine | Ramsey | 62-0057 | 47.0 | 13.4 | 2012 |
| Lac Lavon | Dakota | 19-0446 | 26.7 | 9.8 | 1988 |
| Little Birch | Todd | 77-0089 | 339.7 | 27.1 | 2003 |
| Locke | Wright | 86-0168 | 56.7 | 14.9 | 2011 |
| McCarron | Ramsey | 62-0054 | 29.7 | 17.4 | 2000 |
| McMahon | Scott | 70-0050 | 65.7 | 4.3 | 2007 |
| Mille Lacs | Mille Lacs | 48-0002 | 51,891.3 | 12.8 | 1998 |
| Minnetonka Grays' | Hennepin | 27-013301 | 74.6 | 11.0 | 1987 |
| Minnetonka North Arm | Hennepin | 27-013313 | 127.1 | 17.7 | 1987 |
| Minnetonka Smiths' | Hennepin | 27-013302 | 184.1 | 9.1 | 1987 |
| Minnie-Belle | Meeker | 47-0119 | 240.2 | 14.9 | 2010 |
| Mitchell | Hennepin | 27-0070 | 46.1 | 5.8 | 2002 |
| Mitchell | Sherburne | 71-0081 | 68.6 | 10.1 | 2007 |
| Oscar | Douglas | 21-0257 | 471.7 | 7.6 | 1992 |
| Otter | Anoka | 02-0003 | 127.0 | 6.4 | 1989 |
| Phalen | Ramsey | 62-0013 | 80.0 | 27.7 | 1997 |
| Piersons | Carver | 10-0053 | 108.0 | 12.2 | 1991 |
| Pokegama | Pine | 58-0142 | 601.5 | 7.6 | 2005 |
| Rebecca | Hennepin | 27-0192 | 106.5 | 9.1 | 1989 |
| Riley | Carver | 10-0002 | 119.9 | 14.9 | 1990 |
| Schmidt | Hennepin | 27-0102 | 18.1 | 7.6 | 1990 |
| Somers | Wright | 86-0230 | 61.3 | 6.4 | 2013 |
| South Lindstrom | Chisago | 13-0028 | 184.0 | 10.4 | 2010 |
| Spectacle | Isanti | 30-0135 | 98.2 | 15.7 | 2007 |
| Staring | Hennepin | 27-0078 | 67.6 | 4.9 | 2015 |
| Steiger | Carver | 10-0045 | 67.1 | 11.3 | 2001 |

**Table A2.** *Cont.*

| Lake name | County | Lake ID | Lake Area (hectares) | Maximum Depth (m) | Year of Eurasian Infestation |
|---|---|---|---|---|---|
| Sugar | Wright | 86-0233 | 406.2 | 21.0 | 1990 |
| Swede | Carver | 10-0095 | 175.2 | 3.7 | 2008 |
| Thomas | Dakota | 19-0067 | 16.8 | 2.4 | 2011 |
| Turtle | Ramsey | 62-0061 | 182.1 | 8.5 | 2000 |
| Upper Prior | Scott | 70-0072 | 157.9 | 15.2 | 2000 |
| White Bear | Washington | 82-0167 | 982.5 | 25.3 | 1988 |

**Table A3.** Summary of genetic analyses of lakes surveyed. The number of each taxon identified from samples collected in each lake is presented and the number of distinct genotypes is indicated for each taxon in each lake.

| Lake | County | Counts per Taxon | | | Genotype Counts per Lake | | |
|---|---|---|---|---|---|---|---|
| | | EWM | HWM | NWM | EWM | HWM | NWM |
| Alimagnet | Dakota | | 20 | | | 1 | |
| Auburn | Carver | 24 | | | 1 | | |
| Bald Eagle | Ramsey | 35 | 43 | 50 | 1 | 1 | 3 |
| Ballantyne | Blue Earth | 20 | | | 1 | | |
| Bay | Crow Wing | 14 | | 6 | 1 | | 3 |
| Big Marine | Washington | 12 | | 13 | 1 | | 8 |
| Bone | Washington | | 19 | | | 1 | |
| Cedar | Hennepin | 5 | | | 1 | | |
| Cedar | Wright | | | 20 | | | 6 |
| Christmas | Hennepin | 48 | | 33 | 1 | | 5 |
| Chub | Carlton | 1 | | 19 | 1 | | 1 |
| Cobblestone | Dakota | | 2 | | | 1 | |
| Constance | Wright | 17 | | | 1 | | |
| Coon | Anoka | 11 | 29 | | 1 | 2 | |
| Crooked | Anoka | | 20 | | | 3 | |
| East Rush | Chisago | | 18 | 2 | | 1 | 1 |
| Elmo | Washington | 16 | 23 | | 1 | 1 | |
| Emily | Crow Wing | 2 | | 6 | 1 | | 6 |
| Fish | Dakota | | 20 | | | 1 | |
| Fox | Rice | 20 | | | 2 | | |
| German | Le Seuer | 1 | 9 | 1 | 1 | 5 | 1 |
| Gervais | Ramsey | | | | | | |
| Gilbert Pit | St. Louis | 9 | | | 1 | | |
| Gilchrist | Pope | 20 | | | 1 | | |
| Green | Kandiyohi | 2 | | | 1 | | |
| Ham | Anoka | | 97 | 6 | | 1 | 1 |
| Harriet | Hennepin | 20 | | | 1 | | |
| Howard | Wright | 9 | 10 | 1 | 1 | 6 | 1 |
| Independence | Hennepin | 43 | 44 | | 1 | 1 | |
| Indian | Wright | | 1 | | | 1 | |
| Josephine | Ramsey | | 19 | | | 1 | |
| Lac Lavon | Dakota | | 20 | | | 5 | |
| Little Birch | Todd | 4 | | 15 | 1 | | 6 |
| Locke | Wright | | | | | | |
| McCarron | Ramsey | 21 | 11 | | 1 | 1 | |
| McMahon | Scott | 4 | | | 1 | | |
| Mille Lacs | Mille Lacs | 2 | | 10 | 1 | | 2 |

**Table A3.** *Cont.*

| Lake | County | Counts per Taxon | | | Genotype Counts per Lake | | |
|------|--------|-----|-----|-----|-----|-----|-----|
| | | EWM | HWM | NWM | EWM | HWM | NWM |
| Minnetonka-Grays | Hennepin | | 54 | | | 5 | |
| Minnetonka-North Arm | Hennepin | | 20 | | | 7 | |
| Minnetonka-Smiths | Hennepin | 14 | 37 | 6 | 2 | 10 | 4 |
| Minnie-Belle | Meeker | 1 | | 25 | 1 | | 5 |
| Mitchell | Hennepin | 24 | | 16 | 1 | | 3 |
| Mitchell | Sherburne | 5 | | 34 | 1 | | 3 |
| Oscar | Douglas | 5 | | 15 | 1 | | 5 |
| Otter | Anoka | | 64 | | | 2 | |
| Phalen | Ramsey | 4 | | | 1 | | |
| Piersons | Carver | 19 | | | 1 | | |
| Pokegama | Pine | 5 | | | 1 | | |
| Rebecca | Hennepin | 21 | 8 | | 1 | 1 | |
| Riley | Carver | 21 | | | 1 | | |
| Schmidt | Hennepin | | 62 | | | 2 | |
| Somers | Wright | 2 | | | 1 | | |
| South Lindstrom | Chisago | | 9 | 19 | | 1 | 4 |
| Spectacle | Isanti | 3 | | 22 | 1 | | 4 |
| Staring | Hennepin | 8 | | | 1 | | |
| Steiger | Carver | 20 | | | 1 | | |
| Sugar | Wright | 1 | | 19 | 1 | | 5 |
| Swede | Carver | 13 | | | 1 | | |
| Thomas | Dakota | | 5 | | | 2 | |
| Turtle | Ramsey | 6 | 6 | | 1 | 1 | |
| Upper Prior | Scott | 14 | 10 | | 2 | 2 | |
| White Bear | Washington | 24 | 12 | | 1 | 1 | |

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
