# Peer review of "Factors Influencing the Distribution of Invasive Hybrid (Myriophyllum Spicatum x M. Sibiricum) Watermilfoil and Parental Taxa in Minnesota"

_diversity, doi:10.3390/d12030120_

Round 1

Reviewer 1 Report

I read the paper by Eltawely et al with great interest and found it to be an interesting geographic analysis of the distribution and occurrence of hybrids between invasive Eurasian and native Northern watermilfoil.  I think the results of this analysis are insightful and will be useful for others who are working to understand how/why hybridization occurs, why hybrids appear to be common in some geographic regions and not others, and how hybrids may have differential persistence or resistance to management techniques.  By and large, I think the introduction, methods and results presentation are clear and easy to follow. I appreciate the analyses looking at the environmental and anthropogenic factors that may be related to the distribution of hybrids. I have a few points on presentation of study design and description of study sites for the authors to consider, as well as some concerns about how the results are presented and interpreted in the discussion that I outline below.  

First, how were the study lakes selected? This is important because Figure 1 illustrates that the lakes were clustered around the Twin Cities Metro area, not "statewide" as suggested in the text. The methods on line 94-97 suggest that a county-level stratified sampling approach was taken, but there were many counties state-wide that were not sampled at all. Perhaps this is based on reporting, but my own look at the EDDMapS website suggest that Eurasian watermilfoil has been detected more evenly across the state than reflected in Figure 1. If this has changed since 2017, that's okay, but should be explained clearly here. I also suggest changing the terminology in the paper from "Statewide" and "Greater Minnesota" (which are used interchangably) to some better explanation of the sampling distribution (southeastern MN or something similar).  

Relatedly, I think the analysis separating the lakes within the Metro area from those outside is useful, but referring to the region as "the Metro" is informal and unclear to anyone who is not a resident of the area.  At minimum, this should be revised to "the Metro area" and used consistently throughout, with a definition somewhere in the methods like "The seven counties of the Twin Cities Metro area will hereafter be referred to as "the Metro area""

In the results - I know that Diversity publishes figures in color, but I printed a pdf for review and had difficulty interpreting Figure 1 and 2.  Consider whether shading or color contrast could be used to make these figures intelligible when printed in black and white as well as in color. 

The caption of Figure 1 is where I first became concerned about the fact that the "statewide occurrence" in the text was not statewide on the map.

Lines 286 - 288 - this sentence needs to be rephrased for accuracy - the lakes were not significantly older, the infestations were. The phrase "closer to the nearest documented infestation" makes it sound like the sampled lakes did not also have documented infestations - consider this phrasing throughout.

In the discussion, I had again some questions about the geographic scope, and also found some logical disconnects between the findings and presented here and the results as presented int he prior section

Line 343-344 - "Eurasian was found in most lakes...and widely distributed across the state" - for the first part, make clear that Eurasian was found in most lakes SAMPLED, and for the second part, revise the "across the state" to reflect the scope of sampling presented in Figure 1. 

Lines 404-405, 407-409, 426-427 - this section on clonal spread seems a big jump to me from the data as presented. The results present distribution of hybrids and genotypes, while the discussion here jumps to discussing clonal vs. sexual spread without making the explicit link to how the results were interpreted. Perhaps the results need some clearer interpretation linking the genotypes presented to likely mechanisms of spread, so that these interpretations aren't so surprising when encountered by the reader?  I also suggest that the authors need to be more careful in their language here - instead of "we found to be due to clonal spread" could instead be, "which our analysis suggests are most likely clonally spread because of [describe specific result here]"

Line 419-422 - similarly, the statement "were likely due to within-lake hybrid formation" needs to be more clearly linked to the evidence presented in the results section to make this easy for the reader to interpret and evaluate.

Line 433-435 - could the different results in this study vs. the Sturtevant study be due to sampling error?  E.g., you measured different sets of lakes and found different answers?

Line 438 - seems unlikely that you missed many what?  Incomplete thought here.

Line 442 - does "very few (seven lakes)" refer to the current study, or to the studies described in the prior two sentences?

Line 464-466 - this seems to contradict the statement on lines 419-421 about the likeliness of in-situ hybrid formation in more than half of occurrences.

Author Response

We appreciate the comments of the reviewers and address all of their comments below. In response to reviewer one’s comments we have clarified our lake selection methods and geographic region terminology. Figures 1-2 have been updated to provide better contrast for grayscale printing. In response to reviewer two’s comments we have updated the title to better reflect the content of our manuscript, as well as provided more information in the abstract regarding key factors associated with distribution, and added background information to the introduction regarding invasion history. We also addressed reviewer two’s comments regarding our statistical analysis approaches. We indicate below our responses to specific comments.

Author’s responses to Reviewer 1

  • how were the study lakes selected?
    • The lakes were selected in proportion to country occurrences of listed infestations. We added lines 160-165 to clarify lake selection process
    • We added a new appendix (now Appendix A) for infestation count per county & proportions to illustrate our distribution and renumbered other appendices
  • Statewide occurrence is not statewide
    • See comments and revisions above on distribution of lake sampling effort.Infestations are not completely statewide but we sampled counties statewide in proportion to milfoil occurrence. 
  • Metro: At minimum, this should be revised to "the Metro area" and used consistently throughout, with a definition somewhere in the methods like "The seven counties of the Twin Cities Metro area will hereafter be referred to as "the Metro area""
    • Changed to Metro area and defined Metro area in introduction at lines 99-100
    • At 213-214 clarified “metro” and “greater Minnesota”
    • updated Metro to Metro area throughout paper
  • In the results - I know that Diversity publishes figures in color, but I printed a pdf for review and had difficulty interpreting Figure 1 and 2.  Consider whether shading or color contrast could be used to make these figures intelligible when printed in black and white as well as in color. 
    • We modified figure color/shading schemes for Figures 1 and 2 for better gray scale contrast
    • Changed Figure 1 legend at line 287 and Figure 2 at line 326 to update color scheme for better contrast for grayscale printing
  • Lines 286 - 288 - this sentence needs to be rephrased for accuracy - the lakes were not significantly older, the infestations were. The phrase "closer to the nearest documented infestation" makes it sound like the sampled lakes did not also have documented infestations - consider this phrasing throughout.
    • We changed the wording to indicate infestations were older.
    • We changed wording from the nearest infestation to other documented infestations (line365). We can further alter the wording but think this should be clearer.
  • Lines 404-405, 407-409, 426-427 - this section on clonal spread seems a big jump to me from the data as presented. The results present distribution of hybrids and genotypes, while the discussion here jumps to discussing clonal vs. sexual spread without making the explicit link to how the results were interpreted. Perhaps the results need some clearer interpretation linking the genotypes presented to likely mechanisms of spread, so that these interpretations aren't so surprising when encountered by the reader?  I also suggest that the authors need to be more careful in their language here - instead of "we found to be due to clonal spread" could instead be, "which our analysis suggests are most likely clonally spread because of [describe specific result here]"
    • We added several sentences at the end of the first paragraph of discussion (lines 421-427) to address our assumptions and interpretation of clonal spead and in situ hybridization
    • We added 489-492 to make connection between mechanism of spread and genetic diversity
  • Line 433-435 - could the different results in this study vs. the Sturtevant study be due to sampling error?  E.g., you measured different sets of lakes and found different answers?
    • We added line 534 to address differences in sampling efforts and also added reference to newer results that show a more frequent occurrence. It is also possible that this was a true change over time and we now admit that possibility and suggest that temporal monitoring is needed.
  • Line 438 missed many what?
    • We have clarified that we are unlikely to be missing that many clonal sources for the unique hybrids we found (lines 566-567)
  • Line 442 - does "very few (seven lakes)" refer to the current study, or to the studies described in the prior two sentences?
    • Line 442 we clarified that very few (seven) lakes refers to our study
  • Line 464-466 - this seems to contradict the statement on lines 419-421 about the likeliness of in-situ hybrid formation in more than half of occurrences.
    • Line 488-490 we clarified that stratification of distribution refers to areas within-lake where all three taxa co-occur rather than lake wide

Reviewer 2 Report

The topic of the paper is relevant because it concerns problem of invasion biology and in detail it focuses on hybridization of invasive alien taxon with native one. The presented manuscript is about  Myriophyllum spicatum and its hybrid with M. sibiricum native to North America.

In general, this study was well-designed and done. I have only minor points:

General remark: The main problem in this study is that p. value of 0.10 was used to determine significance. I understand authors why they did it. Majority of interesting results was not significant when 0.05 was applied thus they had to change it. Nevertheless, the obtained results should be treated with caution.

title - it does not reflect the content of the manuscript. Because also M. sibiricum was studied. It should sound like this (proposal): 'Factors influencing invasive alien hybrid Myriophyllum spicatum x M. sibiricum and its congeners in Minnesota' or similarly...

Abstract: some factors controlling distribution of the species should be mentioned. Instead of it authors wrote that 'no evidence was found'

The main problem in this study is that p. value of 0.10 was used to determine significance. I understand authors why they did it. Majority of interesting results was not significant when 0.05 was applied thus they had to change it. Nevertheless, the obtained results should be treated with caution.

Introduction  - a few lines about history of invasion of M. spicatum in the study area will be useful. One of the drivers of invasiveness of a species is residence time therefore in my opinion such information should be included.

Material and methods

I am not sure whether MANOVA test and Tukey HSD test are appropriate here. They requires normal distribution of variables. In case of field data it is difficult. Perhaps PERMANOVA will be better?

Discussion and conclusions - it is well-written. If authors change statistical test then should be changed either.

Author Response

We appreciate the comments of the reviewers and address all of their comments below. In response to reviewer one’s comments we have clarified our lake selection methods and geographic region terminology. Figures 1-2 have been updated to provide better contrast for grayscale printing. In response to reviewer two’s comments we have updated the title to better reflect the content of our manuscript, as well as provided more information in the abstract regarding key factors associated with distribution, and added background information to the introduction regarding invasion history. We also addressed reviewer two’s comments regarding our statistical analysis approaches. We indicate below our responses to specific comments.

Author’s responses to Reviewer 2

  • Introduction  - a few lines about history of invasion of M. spicatum in the study area will be useful
    • added lines 46-48
  • title - it does not reflect the content of the manuscript. Because also M. sibiricum was studied. It should sound like this (proposal): 'Factors influencing invasive alien hybrid Myriophyllum spicatum x M. sibiricum and its congeners in Minnesota' or similarly...
    • revised title lines 2-4
  • Abstract: some factors controlling distribution of the species should be mentioned. Instead of it authors wrote that 'no evidence was found'
    • clarified significant hybrid presence associations at lines 67-69
  • The main problem is that a p. value of 0.1 was used to determine significance
    • There is an extensive literature and debate on the reliance on p-values and the arbitrary nature of a value of 0.05. Field surveys, unlike controlled experiments, often have low power and high variability so the concern is concluding no difference when differences may be real and a 10% error seems acceptable to us. We present all p-values so the reader can draw their own conclusions and believe that is the best approach to presenting results.
  • I am not sure whether MANOVA test and Tukey HSD test are appropriate here. They requires normal distribution of variables. In case of field data it is difficult. Perhaps PERMANOVA will be better?
    • We have added a reference at 254-257 which supports the use of MANOVA with non-normally distributed data. This reference provides explanation for the use of a MANOVA in which the data distribution may be as a result of skewness rather than outliers, as was seen with a few of our variables.